# VISUAL REPRESENTATION ALIGNMENT FOR MULTIMODAL LARGE LANGUAGE MODELS

## ABSTRACT

Multimodal large language models (MLLMs) trained with visual instruction tuning have achieved strong performance across diverse tasks, yet they remain limited in vision-centric tasks such as object counting or spatial reasoning. We attribute this gap to the prevailing text-only supervision paradigm, which provides only indirect guidance for the visual pathway and often leads MLLMs to discard fine-grained visual details from the vision encoder during training. In this paper, we present **VIsual Representation ALignment (VIRAL)**, a simple yet effective regularization strategy that aligns the internal visual representations of MLLMs with those of pre-trained vision foundation models (VFMs). By explicitly enforcing this alignment, VIRAL enables the model not only to retain critical visual details from its own vision encoder but also to complement additional visual knowledge from VFMs, thereby enhancing its ability to reason over complex visual inputs. Our experiments consistently demonstrate performance improvements across all tasks on widely adopted multimodal benchmarks, with gains reaching up to 17.3% and an average improvement of 9.4% over the baseline. Furthermore, we conduct comprehensive ablation studies to validate the key design choices underlying our framework. We believe this simple finding opens up an important direction for the effective integration of visual information in training MLLMs.

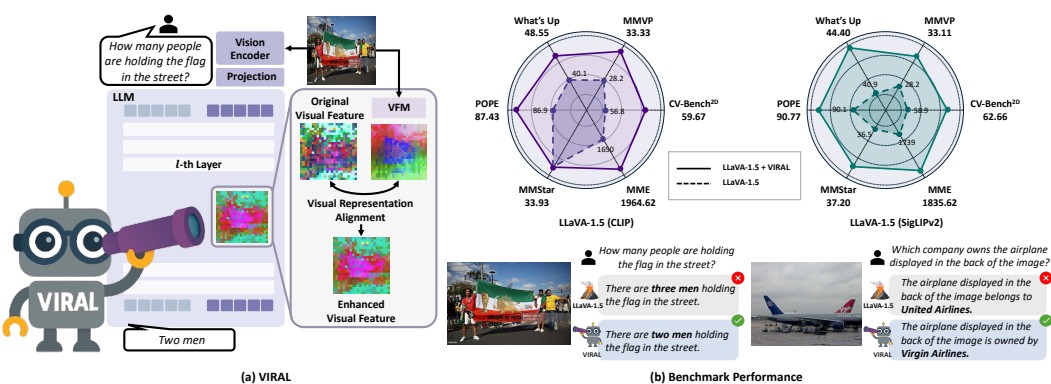

Figure 1: (a) **VIsual Representation ALignment (VIRAL)** introduces an auxiliary regularization objective on the visual pathway, preventing MLLMs from discarding detailed attributes of the input vision encoder during training while incorporating additional visual knowledge from vision foundation models (VFMs). (b) When trained with DINOv2 (Oquab et al., 2023) as the VFM, VIRAL consistently yields more accurate visually grounded responses and achieves substantial improvements over standard baselines (Liu et al., 2023) across diverse vision encoders, including CLIP (Radford et al., 2021) and SigLIPv2 (Tschannen et al., 2025).

## 1 INTRODUCTION

Recent advancements in multimodal large language models (MLLMs) (OpenAI, 2023; Bai et al., 2023a; Team et al., 2023; Chen et al., 2024d), particularly those employing visual instruction tuning

techniques such as LLaVA (Liu et al., 2023), have achieved notable success in diverse multimodal tasks. By connecting pretrained large language models (LLMs) (Touvron et al., 2023; Chiang et al., 2023; Chen et al., 2024d; Bai et al., 2025) with vision encoders (Radford et al., 2021; Chen et al., 2024d; Tong et al., 2024a) through a lightweight vision–language projector, visual instruction tuning enables LLMs to interpret visual context and achieve strong performance across diverse tasks (Chen et al., 2024a; 2025a; Li et al., 2025a).

Despite these successes, numerous studies report persistent limitations in vision-centric tasks such as object counting and spatial reasoning (Tong et al., 2024b; Qi et al., 2025; Yuksekgonul et al., 2022; Ma et al., 2023). Early approaches largely attribute these shortcomings to the visual encoder or the projector. In response, subsequent works have introduced stronger vision encoders (Lu et al., 2024; Li et al., 2024) and more expressive projectors (Liu et al., 2024a; Cha et al., 2024; McKinzie et al., 2024), aiming to supply the language model with richer and more comprehensive visual representations. While they yield notable improvements, approaches that rely solely on more powerful vision encoders or projectors are inherently constrained in scalability and efficiency.

In this paper, we first revisit the conventional training paradigm of visual instruction tuning. Existing MLLMs are predominantly fine-tuned with a language-modeling objective, updating both the LLM and the vision-language projector while concentrating supervision almost entirely on textual outputs (Li et al., 2024; Bai et al., 2023b; Chen et al., 2024d). As a result, visual tokens receive only indirect, language-mediated supervision despite comprising a substantial fraction of the multimodal input. In effect, the visual pathway remains under-supervised, raising a central question: *Is the prevailing multimodal training setup adequate for capturing and preserving visual information?*

We hypothesize that text-only supervision encourages the model to retain only those visual details that immediately aid text prediction, discarding other potentially useful cues. For example, a caption such as *"A photo of a group of people holding a large flag."* provides little incentive to preserve the flag's color, the exact number of people, or their spatial layout—attributes needed for downstream scenarios as in examples shown in Figure 1. In short, text-only supervision aligns visual features with language efficiently (Venhoff et al., 2025; Neo et al., 2024), but does so at the cost of losing the richer and more structured representations provided by the vision encoder.

To validate this hypothesis, we conduct an experiment (see Figure 2) and observe that visual representations trained under exclusive textual supervision rapidly diverge from those produced by the input vision encoder, which we refer to as *visual representation misalignment*. Importantly, we further demonstrate that explicitly preserving alignment with the input vision encoder's representations yields substantial gains in fine-grained visual understanding.

Motivated by these findings, we propose **VIsual Representation ALignment (VIRAL)**, a simple yet effective regularization strategy that directly supervises the visual pathway in MLLMs to prevent the model from discarding fine-grained visual attributes provided by the vision encoder during training. Specifically, we align the internal visual representations of the MLLMs with those of the initial vision encoder using an alignment loss based on cosine similarity. In addition, we further find that this alignment signal is much more effective when provided from stronger vision foundation models (VFMs) (Oquab et al., 2023; Kirillov et al., 2023; Yang et al., 2024; Ranzinger et al., 2024). Since VFMs are trained on vision-centric objectives, they provide rich visual representations that complement language supervision. Therefore, aligning the internal visual representations of MLLMs with those of VFMs likely allows the model to preserve critical visual details while also absorbing additional visual knowledge from VFMs, which in turn enhances its ability to reason over complex visual inputs. Through extensive experiments on widely adopted multimodal benchmarks, we show that VIRAL consistently delivers significant improvements across all tasks.

We summarize our contributions as following:

- We show that, under the visual instruction tuning paradigm, internal visual representations in MLLMs often lose alignment with the rich features produced by vision encoders, leading to the degradation of spatial reasoning capacity due to the loss of fine-grained visual information.

- We propose a novel regularization strategy **VIRAL**, which explicitly aligns MLLM visual representations with features from pretrained VFMs, thereby preventing the loss of fine-grained attributes and enabling richer multimodal understanding.

- Through comprehensive experiments on standard multimodal benchmarks, we show consistent and significant improvements of an average **9.4%** over the baseline. In addition, we conduct extensive ablation studies and analysis to validate our design choices.

## 2 RELATED WORK

**Internal information flows in MLLMs.** Recent studies (Kaduri et al., 2025; Zhang et al., 2025b) have revealed a structured processing hierarchy in MLLMs for vision–language inputs: early layers aggregate global visual context into token embeddings, intermediate layers capture fine-grained spatial features, and later layers integrate multimodal information to facilitate response generation.

Within this hierarchy, the middle layers have been shown to be particularly critical for visual understanding. Jiang et al. (2025) decompose these layers into enrichment and refinement phases, showing that insufficient visual information from earlier stages propagates forward and induces object hallucination. Similarly, Kang et al. (2025) shows that only a small subset of attention heads in the middle layers are pivotal for visual grounding. Consistent with these findings, our analysis of visual representation alignment indicates that the preservation of visual information in the middle layers is strongly linked to spatial reasoning ability, which in turn is crucial for vision-centric tasks.

**Improving visual information in MLLMs.** While recent works have increasingly examined the internal information flow of MLLMs, most prior efforts remain concentrated on the input stage—particularly the use of frozen vision encoders. Improvements at this stage have largely focused on adopting stronger or multiple vision encoders (Kar et al., 2024; Lu et al., 2024; Shi et al., 2024; Azadani et al., 2025) or enhancing efficiency by reducing the overhead of visual tokens (Vasu et al., 2025; Yang et al., 2025; Wen et al., 2025). These advances have proven valuable, yet they primarily address the quality and efficiency of the initial visual representations, with comparatively less attention given to how visual information is processed and propagated once injected into the model. Recent efforts (Wang et al., 2024; 2025) take a step further by advocating direct supervision of visual tokens, but their focus remains on endpoint supervision with less consideration of the internal information flow. Moreover, their reconstruction-based objectives, while effective for preserving low-level fidelity, are less suited for capturing the higher-level semantic abstractions required by complex reasoning tasks (Zhang et al., 2023; Tong et al., 2024a).

In this context, our approach complements these directions by focusing on the internal visual representations—particularly those in the middle layers where fine-grained semantics emerge. By aligning these intermediate features with embeddings from pretrained VFMs, we provide structured supervision that helps preserve semantically meaningful visual content throughout the model.

## 3 PRELIMINARIES

**Multimodal large language models (MLLMs).** MLLMs typically consist of a pre-trained LLM $LM_\theta(\cdot)$ and a vision encoder $V_\psi(\cdot)$, which is connected with a vision-language projector $P_\phi(\cdot)$, where $\theta$, $\psi$, and $\phi$ denote corresponding learnable parameters. To generate answers grounded on both input image and text, the frozen vision encoder $V_\psi(\cdot)$ first extracts patch-level features from an input image $I \in \mathbb{R}^{H \times W \times 3}$ with height $H$ and width $W$ such that $\mathbf{z} = V_\psi(I) \in \mathbb{R}^{N \times D_\mathbf{z}}$, where $N$ and $D_\mathbf{z}$ denote the number of visual tokens and the dimension of the visual features, respectively. Projection modules vary across models—Resampler (Alayrac et al., 2022), Q-Former (Dai et al., 2023), and linear layers (Liu et al., 2023)—with linear layers recently dominating for their simplicity and strong performance. In this case, the linear projector $P_\phi(\cdot)$ maps these visual features into the language model's embedding space, producing a sequence of visual tokens $\mathbf{e}^{\text{img}} = P_\phi(\mathbf{z}) \in \mathbb{R}^{N \times D}$, where $D$ denotes the hidden dimension of the language model. The text sequence is tokenized and embedded into the same embedding space using the language model's token embedding layer, resulting in textual embeddings $\mathbf{e}^{\text{text}} \in \mathbb{R}^{K \times D}$, where $K$ denotes the length of the text tokens. The language model then processes the concatenated multimodal sequence $[\mathbf{e}^{\text{img}}; \mathbf{e}^{\text{text}}] \in \mathbb{R}^{(N+K) \times D}$ and models the causal distribution over the text tokens $\mathbf{e}^{\text{text}}$ as:

$$p_{\theta,\phi}(\mathbf{e}^{\text{text}}_{1:K} \mid \mathbf{e}^{\text{img}}) = \prod_{i=1}^{K} p_{\theta,\phi}(\mathbf{e}^{\text{text}}_i \mid \mathbf{e}^{\text{text}}_{<i}, \mathbf{e}^{\text{img}}). \tag{1}$$

During inference, the language model autoregressively generates text tokens conditioned on the visual representations, the given text prompt, and the previously generated text tokens.

**Training stages of MLLMs.** To enable the language model to incorporate visual information, modern MLLMs typically follow a two-stage training paradigm (Liu et al., 2023; 2024a): a vision–language pretraining stage followed by visual instruction tuning. Both stages share the same language-modeling objective but differ in parameter updates. During vision–language pretraining, only the projector parameters $\phi$ are optimized, while the language model parameters $\theta$ remain frozen. In contrast, visual instruction tuning jointly optimizes both $\phi$ and $\theta$, enabling the language model to adapt more deeply to visual inputs.

It is worth noting that both stages are trained using the same language-centric objective, which is designed to maximize the log-likelihood of the text outputs. Specifically, a language modeling (LM) loss is given by:

$$\mathcal{L}_{\mathrm{LM}} = -\frac{1}{K}\sum_{i=1}^{K}\log p_{\theta,\phi}(\mathbf{e}_i^{\mathrm{text}} \mid \mathbf{e}_{<i}^{\mathrm{text}}, \mathbf{e}^{\mathrm{img}}). \tag{2}$$

## 4 METHODOLOGY

### 4.1 DO MLLMS UNDERGO VISUAL INFORMATION LOSS?

While MLLMs take a substantial number of visual tokens as input, they are typically trained with a text-only language modeling loss applied to the output text tokens. Consequently, all learning signals are mediated through language supervision, and the visual representations $\mathbf{e}^{\mathrm{img}}$ receive no vision-specific supervision, as illustrated in Figure 2-(a). In the absence of explicit visual supervision, we hypothesize that the model learns to prioritize only those visual features that immediately aid textual prediction, often discarding other potentially useful information. This, in turn, causes the internal visual representations to drift away from the rich features produced by the vision encoder—an effect that can undermine performance on tasks requiring complex visual reasoning or grounding.

To empirically validate this hypothesis, we measure the similarity between the internal visual representations of LLaVA (Liu et al., 2024a) and the original visual features $\mathbf{z}$ extracted by its vision encoder (e.g., CLIP (Radford et al., 2021)). We adopt CKNNA (Huh et al., 2024) as a metric to quantify representational similarity.

As shown in Figure 2-(d), similarity to CLIP features drops sharply after the early layers and remains low in deeper layers, indicating that the model's internal visual representations increasingly diverge from the encoder's input features. This trend suggests that, without explicit visual supervision, the model has little incentive to preserve the encoder's rich visual information.

Interestingly, despite the overall decline in alignment, the middle layers show a clear attenuation of this trend, with even slight increase, suggesting that the network implicitly benefits from retaining visual representations at these depths when generating visually grounded answers. This observation aligns with prior analyses of information flow in MLLMs (Zhang et al., 2025b; Kaduri et al., 2025) and is also confirmed by our later layer-wise ablations, which show that leveraging the middle layers for vision-centric tasks shows the largest gains (see Section 5.3).

### 4.2 IS PRESERVING VISUAL INFORMATION BENEFICIAL?

Having observed the mid-layer local increase in representation alignment, we ask whether *explicitly preserving* such visual information is beneficial. Let $\mathbf{e}_\ell^{\mathrm{img}} \in \mathbb{R}^{N \times D}$ denote the visual representations at the $\ell$-th layer of MLLMs. As a direct approach (Figure 2-(b)), we re-inject the projected visual representation $P_\phi(\mathbf{z})$ into an intermediate layer of the language model via a residual path:

$$\mathbf{e}_{\ell,i}^{\mathrm{img}} \leftarrow \mathbf{e}_{\ell,i}^{\mathrm{img}} + P_\phi(\mathbf{z}_i). \tag{3}$$

To isolate the effect of visual information retention without introducing new supervision, the model is trained solely with the original text loss $\mathcal{L}_{\mathrm{LM}}$. Unless otherwise stated, we set $\ell = 16$ in a 32-layer model LLaVA (Liu et al., 2024a), following our analysis that fine-grained visual understanding emerges most prominently in middle layers, supported by later layer-wise ablations (see Section 5.3).

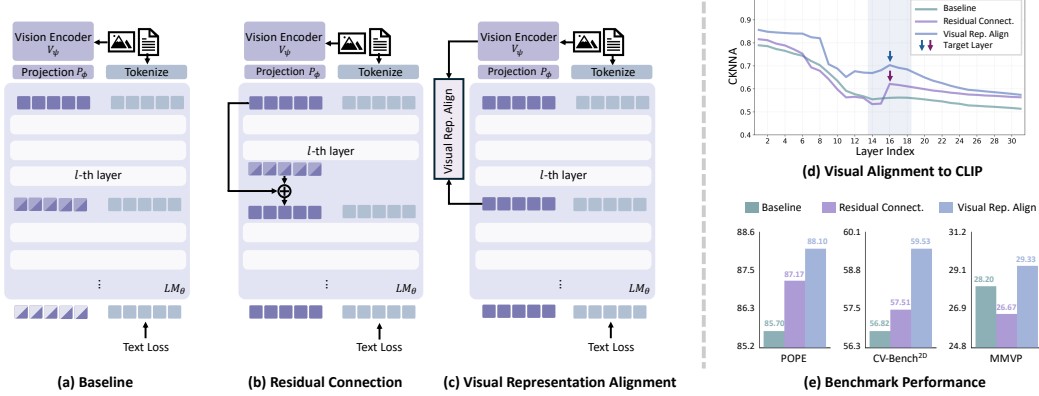

Figure 2: **Re-injecting or aligning visual features improves representation alignment and performance.** (a–c) Comparison of (a) baseline visual instruction tuning (Liu et al., 2023), (b) re-injecting visual features, and (c) visual representation alignment, all applied at the 16th layer. (d) Layer-wise alignment between visual tokens in MLLMs and vision encoder features, measured by CKNNA (Huh et al., 2024), with shaded regions denoting middle layers that are particularly important for visual understanding. (e) Benchmark performance corresponding to (a–c).

As shown in Figure 2-(d), adding the residual connection better preserves the alignment with the encoder's visual features, as indicated by higher CKNNA similarity. Evaluated across standard benchmarks (Figure 2-(e)), this approach shows general improvements over the baseline, supporting the hypothesis that retaining encoder-aligned visual information benefits downstream tasks.

Although residual connection provides general gains, concerns remain that the vision-language projector, $P_\phi(\cdot)$, may not fully preserve the original visual information (Verma et al., 2024; Cha et al., 2024). This raises the question of whether using the encoder's visual representations directly could better preserve visual information. To validate this hypothesis, we explore directions for connecting the raw encoder features directly to the language model in the following part.

### 4.3 VISUAL REPRESENTATION ALIGNMENT FOR MLLMS

**Representation alignment with encoder features.** Beyond residual connection, we further explore a more principled approach, which is to *explicitly* align intermediate visual representations with the encoder features (Yu et al., 2024); see Figure 2-(c). Let $\mathbf{z}$ denote the frozen encoder features from $V_\psi(\cdot)$ and $\mathbf{e}_\ell^{\text{img}} \in \mathbb{R}^{N \times D}$ the visual representations at the $\ell$-th layer of the MLLM. We introduce a learnable projection $P_\pi(\cdot)$ to map $\mathbf{e}_\ell^{\text{img}}$ into the encoder feature space and define the visual representation alignment loss:

$$\mathcal{L}_{\text{VRA}} = -\frac{1}{N} \sum_{i=1}^{N} \text{sim}\left( P_\pi(\mathbf{e}_{\ell,i}^{\text{img}}), \mathbf{z}_i \right), \tag{4}$$

where $\text{sim}(\cdot, \cdot)$ is cosine similarity and gradients do not flow into $\mathbf{z}$. Finally, the total objective augments the language modeling loss with this alignment term:

$$\mathcal{L}_{\text{total}} = \mathcal{L}_{\text{LM}} + \lambda \mathcal{L}_{\text{VRA}}, \tag{5}$$

with $\lambda$ controlling the strength of alignment.

As shown in Figure 2-(d,e), this alignment outperforms residual connection in both CKNNA similarity and multimodal benchmarks. Further analysis on this finding is provided in Appendix B. This shows that constraining intermediate features through an alignment loss offers stronger preservation of fine-grained semantics through explicit regularization, while residual connections offers only weak constraints without enforcing consistency at the feature level.

Despite the general performance boost from retaining encoder-aligned visual information, either by re-injecting projected features or applying visual representation alignment, a notable exception is MMVP (Tong et al., 2024b), which targets cases where CLIP-like features underperform. In this

setting, performance shows only marginal improvement or even a slight drop, suggesting that propagating the encoder's features can also transmit its inductive biases and limitations. These findings raise the question of the *alignment target*: should the model remain tied to the original encoder features $\mathbf{z}$, or be guided toward more informative visual semantics? While aligning to $\mathbf{z}$ helps retain meaningful attributes, its utility is constrained by the encoder's representational capacity.

**From encoder features to other VFMs.** Motivated by this, we adopt stronger vision foundation models (VFMs) as teachers to supervise internal visual representations, providing richer vision-centric targets that complement language supervision. Building on this insight, we propose **VIsual Representation ALignment (VIRAL)**, which aligns intermediate MLLM visual representations with features from a pretrained VFM, thereby preserving richer visual semantics than those available from the encoder alone. Let $\mathcal{E}(\cdot)$ denote a pretrained VFM encoder. Given an input image $I$, the encoder produces target features $\mathbf{y} = \mathcal{E}(I) \in \mathbb{R}^{N \times d}$, where $d$ is the VFM feature dimension. Let $\mathbf{e}_\ell^{\mathrm{img}} \in \mathbb{R}^{N \times D}$ be the MLLM's visual representations at layer $\ell$, and let $P_\pi(\cdot)$ be a learnable projection that maps $\mathbf{e}_\ell^{\mathrm{img}}$ into the VFM feature space. We instantiate the visual representation alignment loss by replacing the encoder target $\mathbf{z}$ in Eq. 4 with $\mathbf{y}$:

$$\mathcal{L}_{\mathrm{VRA}} = -\frac{1}{N} \sum_{i=1}^{N} \mathrm{sim}\Big( P_\pi(\mathbf{e}_{\ell,i}^{\mathrm{img}}), \mathbf{y}_i \Big). \quad (6)$$

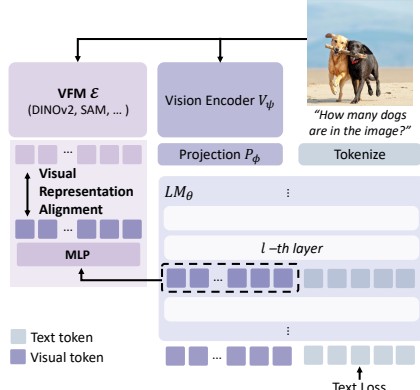

Figure 3: **Illustration of VIRAL.** We align visual pathway representation from MLLMs to strong, informative representations from VFMs to improve the vision understanding performance of MLLMs.

Minimizing $\mathcal{L}_{\mathrm{VRA}}$ regularizes the MLLM's internal visual pathway to align with the VFM. The overall framework is illustrated in Figure 3.

## 5 EXPERIMENTS

### 5.1 EXPERIMENTAL SETTINGS

**Implementation details.** We build on the widely used LLaVA-1.5 (Liu et al., 2024a), leveraging Vicuna-1.5 (Chiang et al., 2023) as the language model with a CLIP vision encoder (Radford et al., 2021). Following its instruction-tuning recipe, we adopt LoRA (Hu et al., 2022) for efficient adaptation as prior work reports that LLaVA-1.5 with LoRA attains comparable performance to full fine-tuning (Liu et al., 2024a). Unless otherwise noted, we use the original LLaVA-665K dataset (Liu et al., 2024a) without any additional data. The visual-representation projector $P_\pi(\cdot)$ is a lightweight three-layer MLP with SiLU activations, and we set $\mathcal{E}(\cdot)$ to DINOv2 as default (Section 5.3).

**Evaluation.** To demonstrate the effectiveness of VIRAL, we evaluate it on widely used benchmarks across three categories: (1) vision-centric tasks requiring spatial reasoning or object counting, including CV-Bench[2D] (Tong et al., 2024a), What's Up (Chen et al., 2025b; Kamath et al., 2023), and MMVP (Tong et al., 2024b); (2) multimodal hallucination detection, using POPE (Li et al., 2023b); and (3) general multimodal understanding, assessed via MME (Yin et al., 2024), MMStar (Chen et al., 2024b). These benchmarks align with goals of our method: improving visual grounding should enhance performance on vision-centric and hallucination-sensitive tasks, while ensuring strong performance on general multimodal benchmarks to preserve overall capability. For evaluation, we report overall accuracy for CV-Bench[2D], MMVP, What's Up, POPE, and MMStar and total score for MME. Additional details on the evaluation settings are provided in Appendix A.

### 5.2 MAIN RESULTS

The results on vision-centric benchmarks, visual hallucination tasks, and general vision–language evaluations are summarized in Table 1. Across identical training settings, the model trained with VIRAL consistently outperforms the baseline—with the largest gains on fine-grained vision-centric

Table 1: **Effects of visual representation alignment.** We compare models trained with and without $\mathcal{L}_{\text{VRA}}$ across various vision encoders and LLM backbones, evaluating them on both vision-centric and general multimodal benchmarks. Our simple regularization, $\mathcal{L}_{\text{VRA}}$, combined with DINOv2 (Oquab et al., 2023), consistently improves performance across all encoders.

| Language Model | Vision Encoder | $\mathcal{L}_{\text{VRA}}$ | CV-Bench$^{2D}$ | MMVP | What's Up | POPE | MMStar | MME |
|---|---|---|---|---|---|---|---|---|
| Vicuna-1.5-7B | CLIP | ✗ | 56.82% | 28.20% | 40.13% | 85.70% | **33.93%** | 1650.21 |
| | | ✓ | **59.67%**(+2.85) | **33.33%**(+5.13) | **48.55%**(+8.42) | **88.32%**(+2.62) | 33.93%(±0.00) | **1694.52**(+44.31) |
| | SigLIPv2 | ✗ | 58.90% | 28.22% | 40.90% | 90.13% | 36.53% | 1738.96 |
| | | ✓ | **62.66%**(+3.76) | **33.11%**(+4.89) | **44.40%**(+3.50) | **90.77%**(+0.64) | **37.20%**(+0.67) | **1835.62**(+96.66) |
| Qwen2.5-7B | CLIP | ✗ | 58.97% | 33.47% | 59.08% | **85.88%** | 39.20% | 1743.56 |
| | | ✓ | **60.50%**(+1.53) | **36.07%**(+2.60) | **63.57%**(+4.49) | 84.92%(-0.96) | **39.67%**(+0.47) | **1765.65**(+22.09) |
| Vicuna-1.5-13B | CLIP | ✗ | 57.51% | 32.30% | 44.44% | 87.12% | 34.47% | 1599.04 |
| | | ✓ | **58.97%**(+1.46) | **37.80%**(+5.50) | **62.26%**(+17.82) | **87.79%**(+0.67) | **37.00%**(+2.53) | **1636.62**(+37.58) |

Table 2: **Ablation study on key design components.** We analyze the effects of (i) different vision foundation models (VFMs) and (ii) alignment target layers, through evaluation on vision-centric and general multimodal benchmarks. All experiments are conducted on the LLaVA-1.5-7B baseline.

| VFM | Layer Index | CV-Bench$^{2D}$ | MMVP | What's Up | POPE | MME |
|---|---|---|---|---|---|---|
| Baseline | | 56.82% | 28.20% | 40.13% | 85.70% | 1650.21 |
| *Ablation studies on different VFMs* | | | | | | |
| DINOv2 | 16 | **59.67%** | **33.33%** | 48.55% | 88.32% | **1694.52** |
| CLIP | 16 | 59.53% | 29.33% | 44.50% | 88.10% | 1548.49 |
| SAM | 16 | 57.58% | 30.27% | **49.84%** | 88.34% | 1648.77 |
| DAv2 | 16 | 58.55% | 28.67% | 47.29% | **88.70%** | 1682.42 |
| RADIO | 16 | 57.59% | 31.80% | 47.35% | 88.52% | 1692.94 |
| *Ablation studies on different target layers* | | | | | | |
| DINOv2 | 4 | 58.55% | 30.67% | 45.05% | 87.68% | 1720.36 |
| DINOv2 | 8 | 58.28% | 27.70% | 48.32% | 88.43% | 1662.67 |
| DINOv2 | 12 | 57.77% | 28.59% | 48.19% | 88.27% | 1648.88 |
| DINOv2 | 16 | **59.67%** | **33.33%** | **48.55%** | 88.32% | 1694.52 |
| DINOv2 | 20 | 55.22% | 27.41% | 48.04% | 88.39% | 1705.97 |
| DINOv2 | 24 | 55.77% | 27.48% | 47.99% | 88.10% | 1740.55 |
| DINOv2 | 28 | 54.87% | 27.19% | 47.82% | **88.56%** | **1755.86** |
| DINOv2 | 32 | 56.12% | 26.52% | 47.60% | 87.32% | 1678.69 |

tasks while retaining strong performance on general multimodal benchmarks—through a simple strategy that aligns intermediate MLLM features with VFM targets to strengthen the visual pathway.

To test whether the observed gains arise only when using CLIP as the vision encoder—by compensating for the limitations of a contrastive-only encoder with visually self-supervised features—we further evaluate SigLIPv2 (Tschannen et al., 2025) as the vision encoder, which is trained with both contrastive and self-supervised objectives. Even with this stronger encoder, our alignment loss yields consistent improvements, showing that the gains stem from the alignment itself. Moreover, to examine whether our method follows a scaling trend and is not confined to a particular language model, we also include results with a scaled-up backbone, comparing Vicuna-1.5-13B against 7B, and with an alternative language backbone, Qwen2.5-7B (Bai et al., 2025). Taken together, these findings highlight a broader principle: regularizing intermediate visual representations is a generally applicable strategy that strengthens MLLMs across vision encoders, scales, and language backbones.

## 5.3 COMPONENT-WISE ANALYSIS

In this ablation study, we conduct a comprehensive analysis of key design choices underlying our framework, focusing on core components: the selection of target visual features and the choice of alignment layer. As in Table 2, we evaluate the impact of each component across five benchmarks (CV-Bench, MMVP, What's Up, POPE, and MME) to validate their respective contributions to the model's performance on vision-grounded tasks. Additional ablation studies on alignment objectives and target layers are provided in Appendix C.

**Vision foundation models.** We begin by identifying the most effective target visual features for enhancing the alignment of internal visual representations within MLLMs. While residual connec-

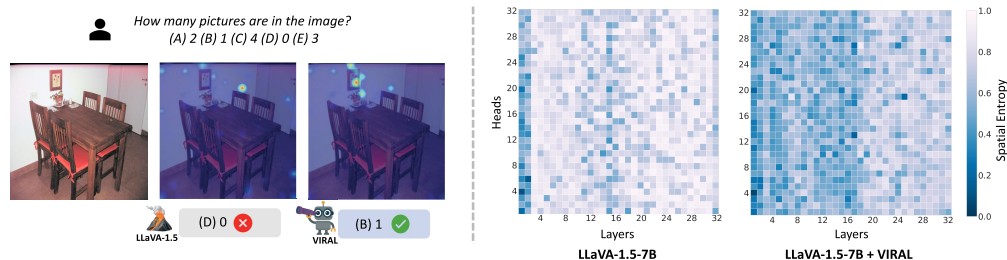

Figure 4: **Analysis of attention.** Qualitative comparison on text-to-image attention maps (left) and quantified spatial entropy of attention across layers and heads (right). Applying visual representation alignment encourages model to attend to more contextually important content, yielding a more focused and structured attention pattern.

tions and alignment with CLIP (LLaVA-1.5's original vision encoder) help improve visual comprehension (Figure 2), their performance on spatial tasks like MMVP is limited—likely due to CLIP's weakness in modeling spatial relations (Yuksekgonul et al., 2022). To address this, we evaluate several stronger vision foundation models (VFMs), including DINOv2 (Oquab et al., 2023), CLIP (Radford et al., 2021), Segment Anything (Kirillov et al., 2023) (SAM), Depth Anything v2 (Yang et al., 2024) (DAv2), and RADIOv2.5 (Heinrich et al., 2025). As shown in Table 2, our analysis confirms that aligning with stronger visual features indeed enhances visual understanding, with DINOv2 and other VFMs demonstrating improved performance compared to CLIP. Results show that DINOv2 consistently emerges as the most effective and versatile, and we thus adopt DINOv2 as the default visual foundation model for all experiments.

**Target layers.** We then analyze alignment at individual target layers to determine the most effective position. As shown in the *target-layers* ablation results in Table 2, we report performance at every 4th layer throughout the network. We observe that performance varies depending on the alignment layer, with the 16th layer of the 32-layer model consistently yielding stronger results across multiple benchmarks. This trend is consistent with prior findings (Zhang et al., 2025b; Kaduri et al., 2025) and our earlier analysis, suggesting that certain intermediate layers in MLLMs are particularly attuned to visual information processing.

### 5.4 ATTENTION ANALYSIS

We analyze the effectiveness of our proposed framework with visual representation alignment in terms of text-to-image attention, as shown in Figure 4 (left). The attention map produced by the $\mathcal{L}_{\mathrm{VRA}}$ trained model exhibits more semantically aligned focus on image regions corresponding to the given textual prompts. To quantify this, we adopt spatial entropy (Batty, 1974), motivated by (Kang et al., 2025), as a metric of attention localization. As shown in Figure 4 (right), LLaVA-1.5-7B exhibits high entropy across layers and heads, reflecting dispersed attention patterns, whereas our model shows consistently lower entropy—particularly at the aligned intermediate layer—indicating more selective and meaningful attention patterns.

### 5.5 ROBUSTNESS ANALYSIS

We investigate whether our representation alignment loss enables MLLMs to better capture spatial relationships. Prior work (Qi et al., 2025) shows that MLLMs often overlook spatial cues, exhibiting only minor performance drops even when the order of visual tokens is randomly permuted (see Appendix A). To assess whether our method makes models more sensitive to such cues, we extract visual features

Table 3: **Robustness to token permutation.** Number of correct predictions out of 788 spatial reasoning tasks in CV-Bench[2D].

| Vision Enc. | $\mathcal{L}_{\mathrm{VRA}}$ | original | patch shuffle | $\Delta$ |
|---|---|---|---|---|
| CLIP | ✗ | 400 | 374 | −26 (6.5%) |
| | ✓ | 414 | 360 | **−54 (13.0%)** |
| SigLIPv2 | ✗ | 374 | 353 | −21 (5.6%) |
| | ✓ | 436 | 353 | **−83 (19.0%)** |

$\mathbf{z} = V_\psi(I)$ from an image $I$, randomly permute the tokens, and feed them into the language model $\mathrm{LM}_\theta(\cdot)$. We then evaluate performance on the spatial reasoning category of CV-Bench[2D]. Table 3 shows that while the text-only baseline undergoes little degradation under permutation, our model

suffers larger drops, reflecting increased sensitivity to spatial structure. This confirms that our loss encourages MLLMs to capture and exploit fine-grained spatial relationships.

## 5.6 QUALITATIVE RESULTS

We qualitatively demonstrate the effectiveness of our proposed approach through detailed analyses of model outputs and internal visual representations. By adopting VIRAL, we observe substantial improvements in performance on vision-centric tasks such as instance counting and understanding spatial relationships. As illustrated in Figure 5, VIRAL correctly answers challenging visual questions related to the number of objects and spatial positioning, whereas the baseline model, LLaVA-1.5-7B, frequently fails.

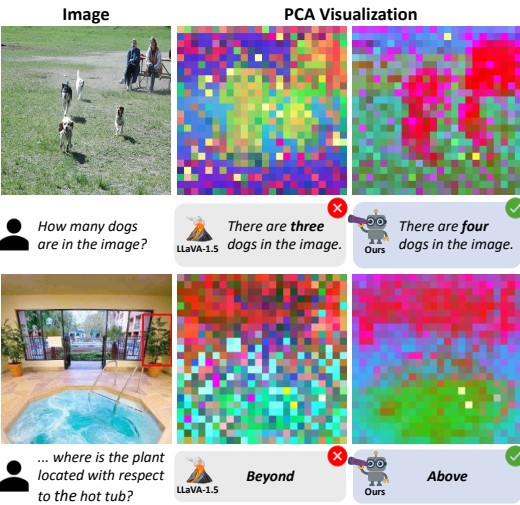

Figure 5: **Qualitative comparison of baseline and VIRAL**. The first column shows the input image–question pairs, and the next two present LLaVA-1.5 and VIRAL results with PCA visualizations and answers. VIRAL yields structured embeddings and correct answers on counting and spatial tasks where the baseline fails.

Furthermore, by aligning internal visual representations with robust vision foundation models (VFMs), the semantic quality of intermediate representations is significantly enhanced. This improvement is clearly evidenced in the PCA visualizations shown in Figure 5. We apply PCA to the visual representations obtained from the 16-th layer of Ours and LLaVA-1.5-7B, where our method yields more structured and semantically coherent embeddings compared to the baseline. These visualizations highlight that our alignment strategy effectively guides the model to preserve critical visual details, thereby facilitating better fine-grained visual comprehension. Additional visualizations are provided in Appendix F.1 and F.2.

## 5.7 TRAINING EFFICIENCY

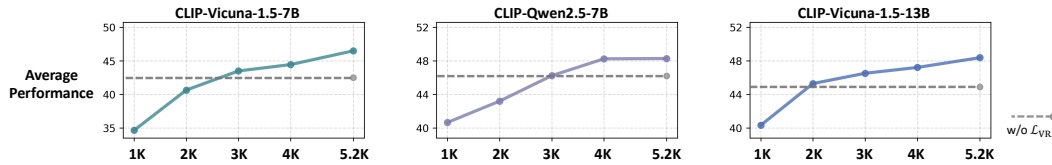

Figure 6: **Training Efficiency.** Performance with $\mathcal{L}_{\text{VRA}}$ (solid) evaluated every 1K steps, averaging accuracies on CV-Bench[2D] and MMVP. Models trained with $\mathcal{L}_{\text{VRA}}$ achieve faster convergence. Dashed lines represent converged performance of baseline.

To further showcase the additional benefits of VIRAL, we evaluated vision-centric benchmarks, including CV-Bench[2D] and MMVP, and averaged the accuracy at every 1K training step from the total of 5.2K training steps of the visual instruction tuning stage in Figure 6. Across three CLIP-based models (CLIP-Vicuna-1.5-7B, CLIP-Qwen2.5-7B, CLIP-Vicuna-1.5-13B), the models trained with VIRALshow that convergence is faster and quickly surpasses the baseline performance in 3K steps. Since our method introduces only about a 3% overhead in total training time, these earlier performance gains may translate into improved scalability.

## 6 CONCLUSION

In this work, we propose VIRAL, a simple yet effective regularization strategy that aligns the internal visual representations of MLLMs with those from pre-trained vision foundation models. Our approach helps preserve fine-grained visual semantics often discarded under text-only supervision, thereby enabling more accurate spatial reasoning and object grounding.

## REPRODUCIBILITY STATEMENT

We detail the training configurations in Section 5.1 and Appendix A. We will also release our code and model checkpoints to ensure reproducibility.

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

# APPENDIX

## A   ADDITIONAL IMPLEMENTATION DETAILS

All experiments in this paper are conducted on four NVIDIA A100 GPUs (40 GB each).

**Vision foundation models.**   We use a diverse set of pretrained VFMs to supervise internal visual representations. DINOv2 (Oquab et al., 2023), CLIP (Radford et al., 2021), and Depth Anything v2 (Yang et al., 2024) (DAv2) are used as patch size 14 models, while RADIO-v2.5 (Heinrich et al., 2025) and SAM (Kirillov et al., 2023) are used as patch size 16 models. To match the 576 visual tokens produced by CLIP-ViT-L/14 at 336×336 resolution in LLaVA-1.5 (Liu et al., 2024a), we adopt the same resolution for patch size 14 models and resize inputs to 384×384 for patch size 16 models. For SAM, which expects 1024×1024 inputs, we pad the interpolated features to 1024×1024 and crop them to the region corresponding to the original image, following AM-RADIO (Ranzinger et al., 2024) to avoid quality degradation.

**Loss function and weighting.**   The cosine similarity $\text{sim}(\mathbf{x}, \mathbf{y})$, as done in previous works, is computed as following $\text{sim}(\mathbf{x}, \mathbf{y}) = \frac{\mathbf{x}^\top \mathbf{y}}{\|\mathbf{x}\|_2 \|\mathbf{y}\|_2}$. To balance the alignment loss $\mathcal{L}_{\text{VRA}}$ with the language modeling loss $\mathcal{L}_{\text{LM}}$, we set $\lambda = 0.5$ by default.

**Benchmark settings.**   To demonstrate the effectiveness of VIRAL, we evaluate it on widely used benchmarks including CV-Bench, MMVP, What's Up, POPE, MME, and MM-Star. We only use the 2D subset of CV-Bench, as 3D tasks are beyond our scope. For simplicity, we report overall accuracy on CV-Bench$^{\text{2D}}$ instead of separately averaging ADE20K and COCO. For MMVP, we follow its standard evaluation protocol using pair accuracy, but for stability, we report the average accuracy over 10 runs. For POPE, we evaluate on COCO following LLaVA and report the average accuracy across the "random" and "popular" subsets. For What's Up, we report the average accuracy between COCO$_{\text{one}}$ and COCO$_{\text{two}}$. For MME, we report MME$^{\text{EN}}$ along with the sum of the perception and cognition categories, and for MM-Star, we follow their standard evaluation protocols.

**Spatial entropy.**   For Figure 4, we compute average spatial entropy over generated text tokens. We use question–answer pairs from (Zhang et al., 2025b), which augment GQA (Hudson & Manning, 2019) with diverse categories and constrain answers to a single word or phrase. Among these, we focus on the Relation category and report the average spatial entropy within this subset.

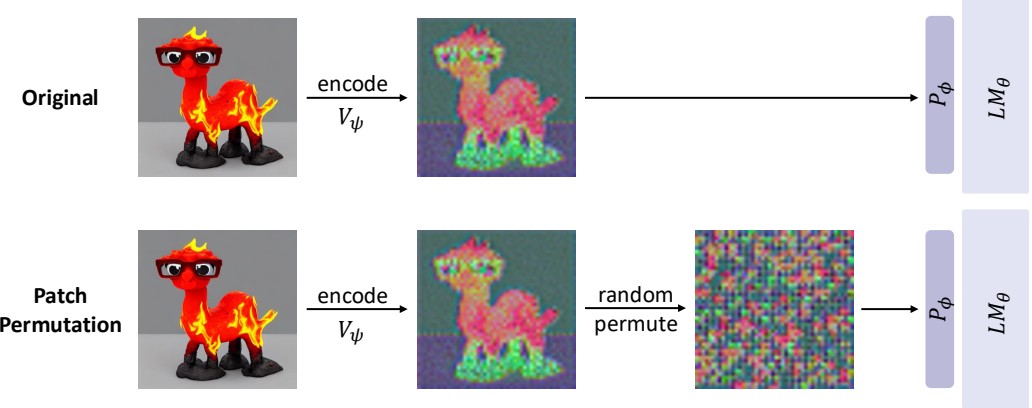

Figure 7: **Visualization of patch random permutation experiments.**

**Patch permutation.**   For our patch permutation experiment, we adopt the analysis pipeline originally proposed in (Qi et al., 2025). Specifically, we begin by extracting image features $z$ from the vision encoder using $z = V_\psi(I)$, where $I$ is the input image. Here, $z \in \mathbb{R}^{N \times H}$, with $N$ denoting the number of visual tokens and $H$ the dimensionality of the vision encoder features. Before processing the vision features $z$ with the vision-language projector $P_\phi(\cdot)$ and language model $LM_\theta(\cdot)$, we apply a random permutation on the order of the visual tokens $N$, which is shown in the visualization of

Figure 7. This makes it extremely difficult to understand the visual attributes of the image, enabling us to evaluate how much the MLLM was understanding and utilizing the visual attributes originally available in the image.

# B  EXTENDED EXPLORATION OF THE PILOT STUDY

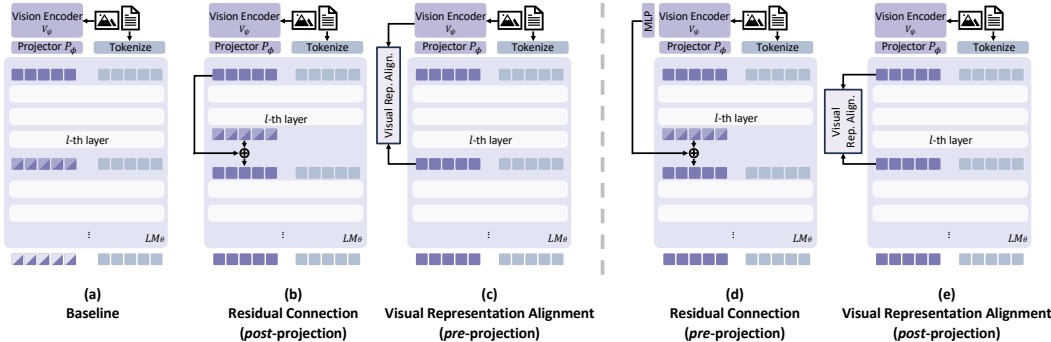

Figure 8: **Extended exploration of the pilot study.**

In Section 4, we demonstrated that MLLMs exhibit progressive visual information loss across layers, and that preserving such information can enhance their visual understanding (Figure 8-(b,c)). In this section, we compare two additional strategies for preserving visual information: a residual connection with the raw encoder feature prior to projection (*pre*-projection) as a direct approach to feature re-injection (Figure 8-(d)), and our proposed visual representation alignment with the projected features (*post*-projection) provided to the language model (Figure 8-(e)).

Table 4: **Benchmark performance of the pilot study.**

|  | POPE | CV-Bench[2D] | MMVP |
|---|---|---|---|
| Baseline | 85.70% | 56.82% | 28.20% |
| (b) | 87.17% | 57.51% | 26.67% |
| (c) | **88.10%** | **59.53%** | **29.33%** |
| (d) | 85.47% | 53.62% | 19.33% |
| (e) | 86.99% | 57.23% | 28.53% |

**Residual connection with *pre*-projection features.**  Our investigation leverages *pre*-projection features—raw encoder features $\mathbf{z}$ prior to the projector—through a direct residual connection to mitigate visual information loss within the language model, where a lightweight adapter $P_{\phi'}(\cdot)$ is employed for dimensional compatibility. As illustrated in Figure 8-(d), we conduct one such experiment that re-injects $\mathbf{z}_i$ into $\mathbf{e}_{\ell,i}^{\mathrm{img}}$ such that

$$\mathbf{e}_{\ell,i}^{\mathrm{img}} \leftarrow \mathbf{e}_{\ell,i}^{\mathrm{img}} + P_{\phi'}(\mathbf{z}_i). \tag{7}$$

However, as shown in Table 4-(d), this approach generally performs worse than the baseline. This is because the raw encoder features, which have not passed through the pre-trained projector, are not sufficiently aligned with language features (Liu et al., 2023), and their direct residual connection consequently disrupts vision–language alignment in the intermediate layers. These findings suggest that incorporating external features into the internal visual pathway of LLMs requires more careful design.

**Visual representation alignment with *post*-projection features.**  Next we further explore aligning the intermediate visual representation with the *post*-projection features, as shown in Figure 8-(e). Here, we follow the same experimental setting as in Section 4.3, while $\mathcal{L}_{\mathrm{VRA}}$ is defined as:

$$\mathcal{L}_{\mathrm{VRA}} = -\frac{1}{N}\sum_{i=1}^{N}\mathrm{sim}\Big(P_{\pi}(\mathbf{e}_{\ell,i}^{\mathrm{img}}),\, P_{\phi}(\mathbf{z}_i)\Big). \tag{8}$$

The results presented in Table 4-(e) indicate that this approach generally improves performance over the baseline on vision-centric benchmarks, yet underperforms compared to leveraging raw features from the vision encoder. This may be attributed to the insufficient preservation of visual information in the *post*-projection features compared to the raw encoder outputs (Verma et al., 2024; Cha et al., 2024; Li et al., 2025b).

## C ADDITIONAL ABLATION STUDIES

Table 5: **Ablation study on key design components.**

| VFM | Ladyer Index | Objective | CV-Bench$^{2D}$ | MMVP | What's Up | POPE | MME |
|---|---|---|---|---|---|---|---|
| Baseline | | | 56.82% | 28.20% | 40.13% | 85.70% | 1650.21 |
| *Ablation studies on different multi-layer targets* | | | | | | | |
| DINOv2 | 16 | Cos. Sim. | **59.67%** | **33.33%** | 48.55% | **88.32%** | **1694.52** |
| DINOv2 | $15-17$ | Cos. Sim. | 59.32% | 28.00% | 47.17% | 87.61% | 1639.72 |
| DINOv2 | $14-18$ | Cos. Sim. | 49.62% | 22.55% | 42.58% | 87.90% | 1444.32 |
| *Ablation studies on different alignment objectives* | | | | | | | |
| DINOv2 | 16 | Cos. Sim. | **59.67%** | **33.33%** | 48.55% | **88.32%** | **1694.52** |
| DINOv2 | 16 | Relation | 58.83% | 26.60% | **49.05%** | 87.58% | 1674.30 |

**Number of target layers.** To investigate the effective number of target layers, we evaluate *multi-layer targets* around the 16th—specifically $\pm 1$ (15–17) and $\pm 2$ (14–18) ranges—and observe that applying alignment solely at the 16th layer achieves the best performance. These findings highlight that aligning visual representations at a specific pathway responsible for visual representation processing, rather than uniformly across multiple layers, is more effective in enhancing the visual understanding capabilities of MLLMs. Based on this observation, we adopt the 16th layer as the default alignment target with DINOv2.

**Alignment objectives.** We investigate the impact of different feature *alignment objectives* during instruction tuning. Specifically, we compare the performance of models trained with a feature relation alignment objective, as a substitute for the proposed direct visual representation alignment loss. Here, the alignment objective is defined as a mean squared error (MSE) loss between the self-similarity matrices of the VFM features and the transformed intermediate representations, which effectively distills the structural relationships among visual features following recent approaches (Zhang et al., 2025a; Bolya et al., 2025). As shown in Table 5, we find that simple cosine similarity-based alignment loss yields higher performance, and adopt it as our default strategy for alignment.

## D COMPARISON WITH OTHER TRAINING OBJECTIVES

Table 6: **Comparison with reconstructive objective.**

| Language Model | Vision Encoder | Objective | CV-Bench$^{2D}$ | MMVP | What's Up | POPE | MMStar | MME |
|---|---|---|---|---|---|---|---|---|
| | | Baseline | 56.82% | 28.20% | 40.13% | 85.70% | 33.93% | 1650.21 |
| Vicuna-1.5-7B | CLIP | ROSS (Default) | 54.24% | 29.73% | 43.57% | 88.19% | **34.73%** | 1648.87 |
| | | ROSS (Middle) | 56.05% | 31.40% | 45.98% | 88.21% | 33.53% | 1647.27 |
| | | VIRAL | **59.67%** | **33.33%** | **48.55%** | **88.32%** | 33.93% | **1694.52** |

We compare our method with ROSS (Wang et al., 2024), which applies a reconstructive objective to the final hidden state of the visual representations. To isolate the sources of improvement, we implement two variants under identical experimental conditions: ROSS (Default), reproducing the original method, and ROSS (Middle), which applies the same objective to an intermediate layer (16th layer as in our configuration for target supervision).

Table 6 reveals several key findings that validate our approach. First, the critical importance of intermediate layer supervision—a contribution of our work—is evidenced by ROSS (Middle) outperforming ROSS (Default), particularly on vision-centric benchmarks. Although both ROSS (Default) and ROSS (Middle) show improvements over the baseline which also shows the importance of providing supervision to the visual pathways, the superiority of ROSS (Middle) over ROSS (Defaults) confirms our hypothesis that supervising visual information flow at strategically chosen intermediate layers, rather than naively at the model's output, yields superior performance gains.

Second, and more fundamentally, our method significantly outperforms both ROSS variants across all benchmarks. This performance gap stems from a crucial distinction in objectives: while ROSS employs reconstruction-based objectives that excel at preserving low-level fidelity, such approaches are inherently less suited for capturing the higher-level semantic abstractions required by complex reasoning tasks (Zhang et al., 2023; Tong et al., 2024a). In contrast, our direct alignment with pretrained vision foundation models provides richer semantic supervision that better bridges the vision-language gap.

These results demonstrate that our method's superiority arises from two synergistic contributions: (1) the strategic placement of supervision at critical intermediate layers, and (2) the use of semantically-rich alignment signals from vision foundation models rather than reconstruction-based objectives. Together, these design choices enable more effective visual representation learning for multimodal understanding.

## E  APPLICABILITY OF VIRAL TO OTHER MLLM ARCHITECTURES

Recent MLLMs employ various strategies to handle inputs with dynamic resolutions, including dynamically adjusting the sequence length of visual tokens (Bai et al., 2023b) or dividing high-resolution images into independently encoded tile grids (Liu et al., 2024b; Chen et al., 2024c). The latter approach preserves the original image resolution and is commonly adopted to capture fine-grained visual details.

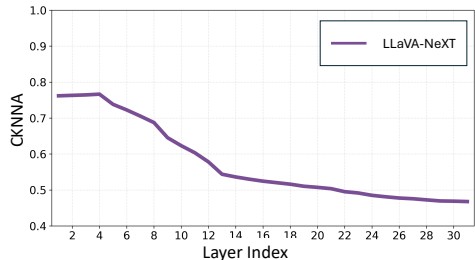

Figure 9: **Visual alignment of LLaVA-NeXT (Liu et al., 2024b).** Layer-wise alignment between visual tokens in MLLM and vision encoder features, measured by CKNNA and averaged across representations from tiled image splits.

To investigate whether VIRAL can be applied to such recent MLLM paradigms, we examine if our core motivation—mitigating vision information loss—remains relevant within this tiled image processing strategy. Figure 9 demonstrates a decline in alignment scores between input visual features and layer-wise visual representations across the layers in LLaVA-NeXT (Liu et al., 2024b), as measured using CKNNA (Huh et al., 2024). This shows similar patterns observed in Figure 2(d), suggesting that VIRAL can be applied orthogonally to such techniques and has the potential to similarly enhance fine-grained visual understanding in MLLMs designed for dynamic resolution handling.

## F  ADDITIONAL VISUALIZATIONS AND RESULTS

### F.1  LAYER-WISE INTERNAL REPRESENTATIONS

We present PCA visualizations of the intermediate visual representations from all layers of LLaVA-1.5-7B and VIRAL in Figure 10, enabling a layer-wise comparison of their representational structures. A qualitative comparison with the baseline reveals that visual representation alignment regularizes the MLLM's internal visual features, leading to more semantically coherent and structured representation, especially in the middle and later layers where meaningful vision understanding emerges.

We present PCA visualizations of the intermediate visual representations from all layers of LLaVA-1.5-7B and VIRAL in Figure 10, enabling a layer-wise comparison of their representational structures. A qualitative comparison with the baseline reveals that visual representation alignment regularizes the MLLM's internal visual features, leading to more semantically coherent and structured representation, especially in the middle and later layers where meaningful vision understanding emerges.

### F.2 VISUAL REPRESENTATIONS WITH DIFFERENT VFMS

In addition to Figure 5, we qualitatively present in Figure 11 PCA visualizations of how internal visual representations evolve when aligned with different VFMs. Compared to the baseline representation from LLaVA-1.5-7B, VFM features exhibit more semantically structured organization. Aligning the MLLM's internal representations with these VFM features distills such structure, enabling the model to refer to enhanced and more coherent visual representations.

### F.3 ATTENTION MAP VISUALIZATIONS

In Figure 12, we provide visualizations of text-to-image cross-attention maps in the MLLM to qualitatively support the attention analysis from the main paper. Compared to the baseline, the model trained with our method exhibits improved attention behavior by focusing more accurately and locally on regions relevant to the given multimodal context. This observation aligns well with the spatial entropy analysis in Figure 4, where models trained with visual representation alignment show more focused and discriminative attention patterns.

## G LIMITATIONS

While our method demonstrates general improvements across vision encoders, model scales, and language backbones (Table 1), several considerations remain. First, since VFMs generally produce representations aligned with the spatial grid of the original image, our alignment relies on projection modules that preserve this structure (e.g., linear projection layers, the de facto choice in current MLLMs (Li et al., 2024; Chen et al., 2024d; Bai et al., 2023b; Lu et al., 2024)). Architectures such as Resampler (Alayrac et al., 2022; Wang et al., 2023) or Q-Former (Dai et al., 2023; Li et al., 2023a) disrupt this grid, making our approach less directly applicable. In the same vein, effective alignment also requires that the resulting grid be resolution-adjustable so that the number of visual tokens matches those expected by the MLLM. Also, because our alignment strengthens the semantic utility of each vision token and their relationships, approaches that rely on token pruning to exploit redundancy (Vasu et al., 2025; Wen et al., 2025) may yield reduced benefits when combined with our method. Finally, while our experiments—as well as prior studies—indicate that the middle layers of MLLMs are primarily responsible for fine-grained information, this behavior may not hold universally as more diverse architectures continue to emerge.

## H USE OF LARGE LANGUAGE MODELS

In accordance with the ICLR 2026 submission policy, we disclose that Large Language Models were used to assist in grammar correction and polishing of the writing in this paper.

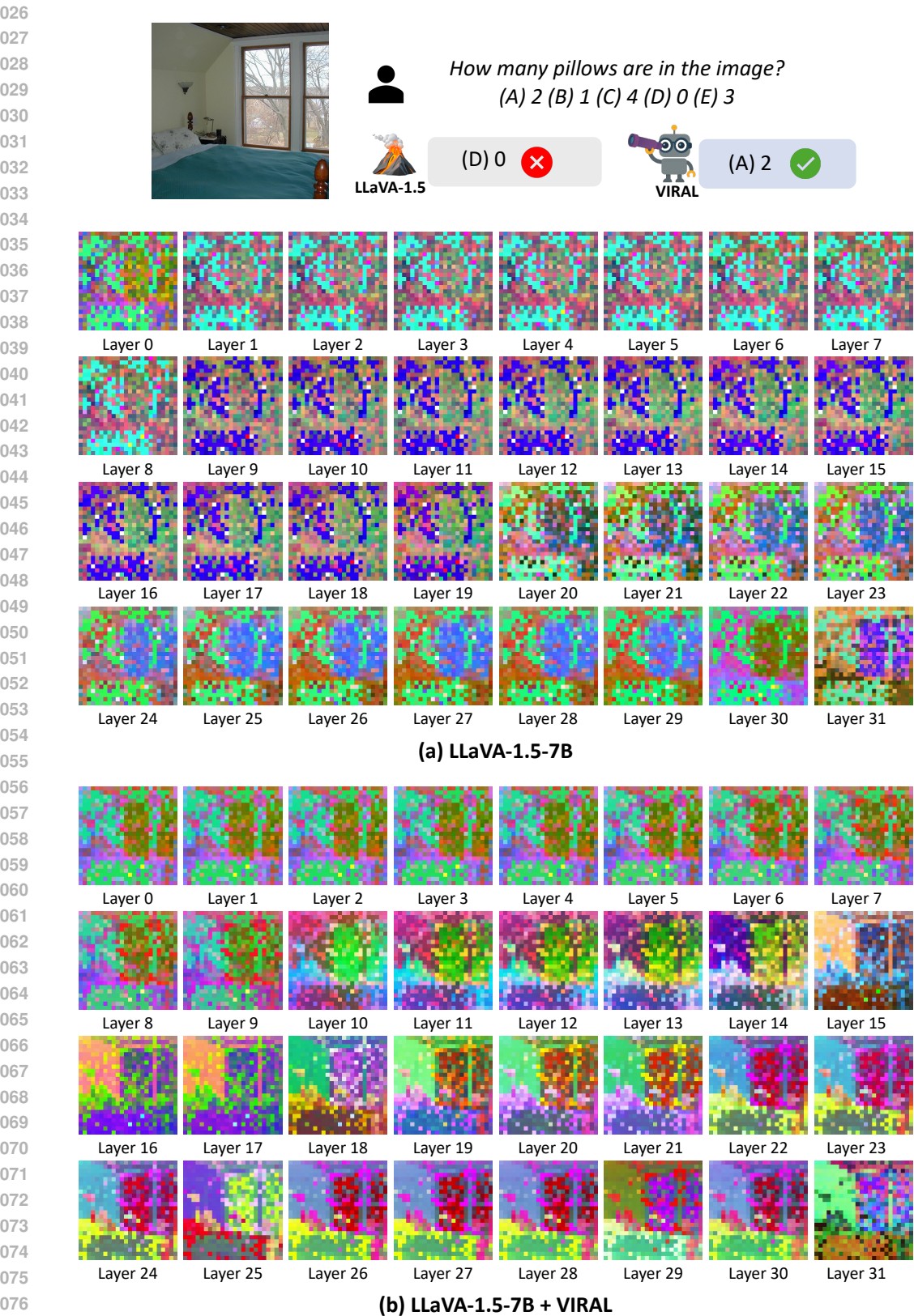

Figure 10: **Layer-wise PCA visualizations of visual representations** from (a) LLaVA-1.5-7B and (b) VIRAL (Ours).

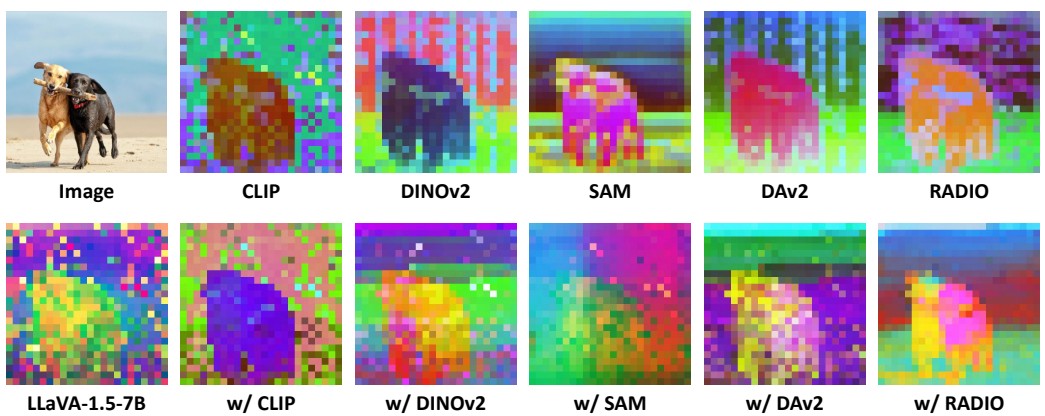

Figure 11: **PCA visualizations of 16th layer visual representations** aligned with different VFMs: CLIP (Radford et al., 2021), DINOv2 (Oquab et al., 2023), SAM (Kirillov et al., 2023), DAv2 (Yang et al., 2024), and RADIO (Heinrich et al., 2025).

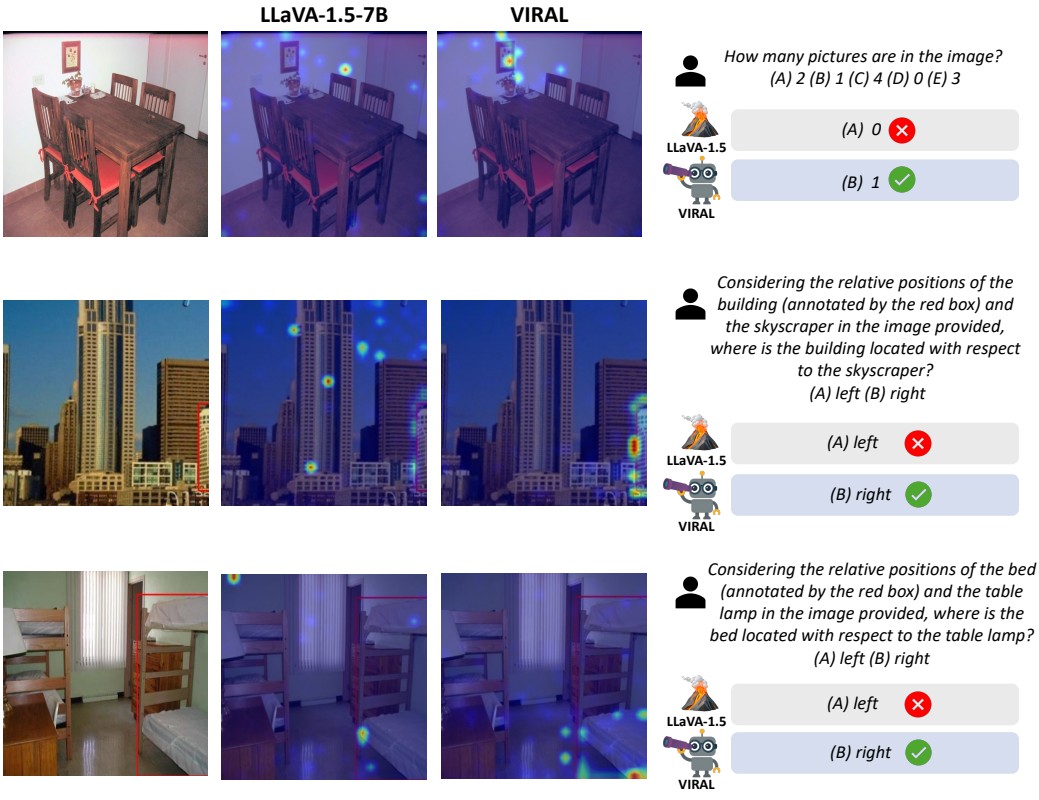

Figure 12: **Cross-attention map comparison for vision centric tasks.**

