# OpenReview forum: "Visual Representation Alignment for Multimodal Large Language Models"
_ICLR.cc/2026/Conference — ICLR 2026 Conference Withdrawn Submission_

### Official Review · Reviewer_j4xG · 2025-10-29

**Soundness:** 2
**Presentation:** 2
**Contribution:** 2
**Rating:** 4
**Confidence:** 4

**Summary:**

This paper introduces the Visual Representation Alignment (VIRAL) to align the internal visual representations of MLLMs with vision foundation models. VIRAL can not only retain critical visual details but also complement additional visual knowledge from VFMs. Experiment results demonstrate VIRAL achieves performance improvement across multiple downstream benchmarks.

**Strengths:**

This work is well-motivated and employs DINOv2 to supervise the visual features of specific layers in LLaVA-1.5, achieving commendable results across multiple benchmarks.

**Weaknesses:**

1. The number of test benchmarks is too limited to comprehensively evaluate the enhancements this method brings to VLM models.
2. Introducing a new model for supervising visual features will increase computational costs. The absence of validation and evaluation on more state-of-the-art VLM platforms makes it difficult to assess the generalizability of this method.

**Questions:**

1. The baselines compared in this work are somewhat outdated. Why are more recent models, such as LLaVA-Next and LLaVA-Onevision, not included in the comparison?
2. This paper only conducts ablation on layer indices using DINOv2. However, it remains unclear whether the optimal layer indices would significantly differ when using different visual encoders.

---

### Official Review · Reviewer_RSrQ · 2025-10-31

**Soundness:** 3
**Presentation:** 2
**Contribution:** 2
**Rating:** 4
**Confidence:** 3

**Summary:**

This paper introduces VIRAL, a regularization strategy that aligns intermediate visual representations in MLLMs with features from pretrained visual foundation models. The approach mitigates the loss of fine-grained visual details caused by text-only supervision, thereby enhancing spatial reasoning and visual understanding.

**Strengths:**

1. The phenomenon studied is interesting and worth investigating. The paper provides the first systematic analysis of visual representation degradation under text-only supervision and proposes an elegant alignment strategy that effectively improves visual comprehension in MLLMs.
2. The proposed method demonstrates strong and consistent improvements across multiple MLLM backbones and benchmarks, with minimal additional training cost.
3. The paper is well-written and easy to follow.

**Weaknesses:**

1. While the proposed alignment strategy shows promising results, the paper does not include comparisons with recent multi-encoder MLLMs (e.g., Eagle [1]) that explicitly fuse complementary vision encoders for fine-grained perception.
2. As many modern MLLMs employ tiled high-resolution inputs (e.g., LLaVA-NeXT) to enhance fine-grained visual reasoning, a quantitative study of VIRAL under such settings would help clarify its incremental benefit.

[1] Eagle: Exploring the design space for multimodal llms with mixture of encoders, ICLR 2025

**Questions:**

It would be helpful to know whether the proposed method has been evaluated for its generalization to video-based multimodal tasks, or if the authors believe the framework could be readily extended to such settings.

---

### Official Review · Reviewer_shHW · 2025-10-31

**Soundness:** 2
**Presentation:** 3
**Contribution:** 1
**Rating:** 2
**Confidence:** 4

**Summary:**

This paper proposes VIRAL (Visual Representation Alignment), a regularization strategy that aligns intermediate visual representations in MLLMs with features from pre-trained vision foundation models (VFMs). The authors argue that text-only supervision causes MLLMs to discard fine-grained visual details, and explicit alignment with VFM features can preserve this information, improving performance on vision-centric tasks.

**Strengths:**

- The experiments show consistent performance improvements, demonstrating solid empirical gains across multiple benchmarks.
-  The observation that text-only supervision leads to visual information loss is interesting and the progression from identifying the problem in Sec. 4.1 to proposing solutions is logical.

**Weaknesses:**

- The main concern is the limited novelty and incremental contribution. The core idea of aligning intermediate representations using cosine similarity is standard practice in knowledge distillation and representation learning, and using pre-trained vision models to supervise morden model training is not new, see [1].
- Insufficient analysis of key design choices. For example, why does DINOv2 work better than other VFMs? The paper doesn't provide deep analysis beyond showing empirical results.

[1] Representation Alignment for Generation: Training Diffusion Transformers Is Easier Than You Think. ICLR 2025.

**Questions:**

Why increased sensitivity to token permutation is desirable?

---

### Official Review · Reviewer_By5h · 2025-11-02

**Soundness:** 3
**Presentation:** 3
**Contribution:** 2
**Rating:** 4
**Confidence:** 4

**Summary:**

This paper proposes VIRAL, a regularization strategy aligning the internal visual representations of MLLMs with pre-trained vision foundation models (VFMs). VIRAL achieves up to 17.3% and on average 9.4% improvement across benchmarks, with ablation studies confirming its effectiveness. The method offers a new direction for integrating fine-grained visual information into MLLMs.

**Strengths:**

1. Clearly targets the core issue of visual detail loss under text-only supervision and effectively enhances visual reasoning.
2. Demonstrates strong generality across various encoders and models with consistent benchmark gains.
3. Adds only ~3% training cost while accelerating convergence, surpassing baselines within 3K steps.

**Weaknesses:**

1. Limited benchmark coverage, lacking broader validation and analysis of potential trade-offs with language ability.
2. No systematic comparison with other integration methods such as DeepStack, making relative advantages unclear.
3. Insufficient theoretical explanation of cross-modal alignment, with little analysis of feature-space consistency or gradient flow.
4. Depends on spatial-grid projection modules, limiting adaptability to Resampler or Q-Former structures.
5. May reduce pruning efficiency when combined with visual token compression techniques.

**Questions:**

1. Could the evaluation be expanded to more datasets, and could the trade-offs with language performance be analyzed?
2. Could comparisons with DeepStack be added to further clarify the advantages of this method?

---

### Author Response · Authors · 2025-11-27
**General response**

We thank all reviewers for their time and effort in reviewing our manuscript. We appreciate the reviewers' acknowledgment of our method's consistent improvements across multiple benchmarks. We also thank the reviewers for recognizing that our analysis of the limitations in text-only supervision in existing visual instruction tuning stages for MLLMs presents a novel and valuable perspective. We maintain confident that our work makes significant contributions to the field:

(1) We are the first to systematically analyze and demonstrate the visual information degradation that occurs under text-only supervision in visual instruction tuning, where we introduce CKNNA as an effective metric to reveal how the visual information loss happens throughout the layers of MLLMs;

(2) We propose a efficient solution through visual representation alignment that effectively addresses this limitation, which is an effective step towards building more foundational vision-language models, substantially improving fine-grained visual understanding and spatial reasoning tasks.

While we value all the feedback received, we found that parts of our contributions were not fully conveyed, with a few points revisiting minor limitations we had already noted. With this in mind, we have decided to withdraw the submission to improve clarity and incorporate the helpful suggestions for a future submission. Nevertheless, we respectfully note a few points that may benefit from clarification:

**(1) Regarding the comparison with REPA [1]:**
While VIRAL shares surface-level similarities with REPA [1] in terms of using DINOv2 features and cosine similarity for alignment, the fundamental motivations differ substantially. REPA optimizes Diffusion Transformers by boosting DiT’s inherently discriminative features, whereas our work addresses a critical failure mode in MLLMs, which is the progressive loss of visual information under text-only supervision. Our analysis demonstrates that this visual information loss results in degraded spatial reasoning and fine-grained visual understanding, and our solution effectively mitigates the loss of visual information, leading to substantially improved spatial reasoning and fine-grained visual understanding performance. This represents a fundamentally different problem domain and solution paradigm.

**(2) Regarding the comparison with multi-encoder MLLMs:**
While VIRAL and prior approaches share some similarities in terms of using features from multiple vision foundation models, our central claim is that training MLLMs under text-only supervision inherently leads to visual information degradation throughout the network, and the proposed VRA loss directly mitigates this effect by preserving visual information across layers using features from vision foundation models. While multi-encoder MLLMs provide stronger visual features at the input level, when trained with text-only supervision, we believe a similar visual information loss will occur as observed in our experiments with stronger vision encoders (CLIP → Siglipv2), and LLaVA-Next in Appendix E. For these reasons, we would like to clarify that VIRAL stands out as a distinct work that points out and address the issue of text-only supervision in MLLMs, which provides complementary advantages to multi-encoder MLLMs offering a universal improvement that can be integrated with various architectural designs.


[1] Representation Alignment for Generation: Training Diffusion Transformers Is Easier Than You Think. ICLR 2025.

---

### Note · Authors · 2025-11-27

I have read and agree with the venue's withdrawal policy on behalf of myself and my co-authors.